# Gill Mucus and Gill Mucin *O*-glycosylation in Healthy and Amebic Gill Disease-Affected Atlantic Salmon

**DOI:** 10.3390/microorganisms8121871

**Published:** 2020-11-26

**Authors:** John Benktander, János T. Padra, Ben Maynard, George Birchenough, Natasha A. Botwright, Russel McCulloch, James W. Wynne, Sinan Sharba, Kristina Sundell, Henrik Sundh, Sara K. Lindén

**Affiliations:** 1Department of Medical Biochemistry and Cell Biology, Institute of Biomedicine, Sahlgrenska Academy, University of Gothenburg, Box 440, Medicinaregatan 9A, 405 30 Gothenburg, Sweden; john.benktander@gu.se (J.B.); janos.tamas.padra@gu.se (J.T.P.); george.birchenough@gu.se (G.B.); sinan.sharba@gu.se (S.S.); 2CSIRO Agriculture and Food, Castray Esplanade, Hobart, TAS 7000, Australia; ben@nextlevelkayaking.com.au (B.M.); james.wynne@csiro.au (J.W.W.); 3CSIRO Agriculture and Food, 306 Carmody Road, St Lucia, QLD 4067, Australia; Natasha.Botwright@csiro.au (N.A.B.); russell.mcCulloch@csiro.au (R.M.); 4Department of Biological and Environmental Sciences, University of Gothenburg, 405 30 Gothenburg, Sweden; kristina.sundell@bioenv.gu.se (K.S.); henrik.sundh@bioenv.gu.se (H.S.)

**Keywords:** glycosylation, amebic gill disease, mucosal immunology, mucin, mucus, gill, *Neoparamoeba perurans*, parasite, Atlantic salmon

## Abstract

Amoebic gill disease (AGD) causes poor performance and death in salmonids. Mucins are mainly comprised by carbohydrates and are main components of the mucus covering the gill. Since glycans regulate pathogen binding and growth, glycosylation changes may affect susceptibility to primary and secondary infections. We investigated gill mucin *O*-glycosylation from Atlantic salmon with and without AGD using liquid chromatography–mass spectrometry. Gill mucin glycans were larger and more complex, diverse and fucosylated than skin mucins. Confocal microscopy revealed that fucosylated mucus coated sialylated mucus strands in ex vivo gill mucus. Terminal HexNAcs were more abundant among *O*-glycans from AGD-affected Atlantic salmon, whereas core 1 structures and structures with acidic moieties such as *N*-acetylneuraminic acid (NeuAc) and sulfate groups were less abundant compared to non-infected fish. The fucosylated and NeuAc-containing *O*-glycans were inversely proportional, with infected fish on the lower scale of NeuAc abundance and high on fucosylated structures. The fucosylated epitopes were of three types: Fuc-HexNAc-R, Gal-[Fuc-]HexNAc-R and HexNAc-[Fuc-]HexNAc-R. These blood group-like structures could be an avenue to diversify the glycan repertoire to limit infection in the exposed gills. Furthermore, care must be taken when using skin mucus as proxy for gill mucus, as gill mucins are distinctly different from skin mucins.

## 1. Introduction

All mucosal epithelia are covered by a continuously secreted mucus layer and mucins are the main components of the mucus [1,2]. Mucins are comprised of 50–90% carbohydrate in the form of *O*-glycans, and substantial differences in glycosylation have been demonstrated in epithelia within and between species [3,4]. However, the inter-individual variation in glycosylation between Atlantic salmon (ATS, *Salmo salar*) is very low compared to that of pigs and humans [3,4,5,6,7]. Since the large inter-individual variation in humans is considered a population-based defense against infection, the low variation in ATS glycosylation may indicate that they are vulnerable to infection at a population level [3,4]. Furthermore, glycosylation can change with infection and inflammation [5,8]. Since glycans regulate pathogen binding and growth [8,9], glycosylation changes may affect susceptibility to both primary and secondary infections.

Skin *O*-glycans from ATS have been identified as being of core type 1, 2, 3 and 5 and relatively short (2–6 monosaccharides) [4]. The ATS mucin *O*-glycans are highly sialylated (~90% sialylation), with *N*-acetylneuraminic acid (NeuAc) being the most abundant sialic acid found both in skin and intestine, while *N*-glycolylneuraminic acid (NeuGc)- and keto-deoxynononic acid (Kdn)-containing structures were only detected on skin mucins [3,4]. The gill mucin glycosylation has not been studied in detail previously.

Amoebic gill disease (AGD) is caused by the amphizoic marine parasite *Neoparamoeba perurans*, and primarily affects salmonid species such as ATS and rainbow trout (*Oncorhynchus mykiss*) [10]. AGD is a major health issue for ATS aquaculture in Australia, but has also been found in farmed fish in Ireland, Scotland, Norway and Chile [11]. In Australia, AGD is treated by freshwater bathing, which may need to be repeated a number of times during the production cycle. This can be a problem due to limited access to freshwater and production cost increases [12]. While grossly AGD is characterized by raised multifocal lesions on the gill surface [13], histologically, AGD causes fusion of the lamellae, epithelial desquamation and oedema, epithelial hyperplasia and interlamellar vesicle formation [14]. Increases in mucus cell numbers at the distal tips of non-hyperplastic secondary lamellae have been described in ATS with naturally occurring AGD together with a tendency for increased levels of periodic acid-schiff (PAS)-positive mucins [14]. AGD may also change the mucus of other epithelia not primarily affected by the AGD itself, since the viscosity of skin mucus from salmonids affected by AGD is lower than that of uninfected fish [15].

The aims of the present study were to characterize the ATS gill mucin *O*-glycome, compare it with the skin and gastrointestinal *O*-glycomes, explore gill mucus structure in relation to mucin glycosylation and investigate how AGD affects the gill mucin *O*-glycome.

## 2. Materials and Methods

### 2.1. Swedish Freshwater Atlantic Salmon

The use of Swedish ATS (ATS-SE) was approved by the Ethical Committee for Animal Experiments in Gothenburg, Sweden under license #177/2013. ATS-SE were raised in fresh water (FW) at Långhults Lax AB, Långhult, Sweden, from wild parents from Göta Älvslax parent fish of the Säveån strain and transported to the experimental animal facilities at the Department of Biological and Environmental Sciences, University of Gothenburg. The fish were kept in 500 L tanks in recirculating FW supplemented with 10% salt water (salinity of 2–3‰) at a flow rate of 8.5 L/min with a temperature of 10 °C, fed a commercial dry pellet (Nutra Olympic 3 mm; Skretting Averøy Ltd., Averøy, Norway) once daily to satiety and exposed to a simulated natural photoperiod for at least two weeks before sampling. Fish harvested for mucin isolation (250–300 g) were sedated in metomidate (12.5 mg/L) and killed with a blow to the head. For mucin isolation, mucus was gently scraped from the entire fish skin and distal intestine using glass microscope slides. The pyloric caeca were placed in liquid nitrogen for grinding using a mortar and pestle. Gills were cut from the gill arches and frozen. Fish for the ex vivo imaging experiments (~30 g) were euthanized in buffered MS-222 (200 mg/L; Sigma, St. Louis, MO, USA). For ex vivo imaging of gill mucus, the gills were excised and immediately used.

### 2.2. Australian Marine Atlantic Salmon

Australian ATS (ATS-AU) were originally of wild type from River Philip (NS, Canada) but have been bred in captivity in Tasmania since the 1960s and are considered a single strain. ATS-AU were spawned and hatched in Tasmania, transported to the CSIRO Bribie Island Research Centre (Queensland, Australia) at the fry stage and reared in a 5000 L freshwater recirculation system under the approved CSIRO Queensland AEC (CQ AEC) application 2017-35. All fish for this study, control (naïve) and AGD-affected fish, at approximately 250 g were adapted to the marine environment by increasing the photoperiod to a 24 h light regime to trigger smolting. After 4 weeks, the photoperiod was changed to 12:12 h light:dark and the salinity gradually increased from 3‰ to 35‰ overnight. Animals were then transferred to a 5000 L seawater flow-through system. The salinity was maintained at 32–35 ppt, flow rate of ~4–5 L/min with a temperature of 16 °C. Fish were fed a commercial dry pellet via an autofeeder (Arvotec-Oy) (Spectra SS 4 mm, Skretting Ltd., Tasmania, Australia) to satiety over the course of the day. Under Australian regulations, the sharing of tissues between applications is encouraged where the care and euthanasia has already been approved and monitored as part of an earlier AEC-approved activity (CQ AEC EX 2020-05). Therefore, gill samples from naïve (control) (CQ AEC application 2018-01) and AGD-affected fish (CQ AEC application 2017-36) were obtained from trials running for the same duration, in parallel, with the same cohort of smolted fish.

The control fish (*n* = 20) were transferred to a flow-through seawater system and housed in a 1000 L tank with flow rate of ~8–10 L/min. The system is provided with filtered, ozonated and UV-disinfected seawater brought into the facility via offshore spear pumps. Water parameters maintained include salinity (~35 ppt), pH 7.8, temperature (16 ± 1 °C) and dissolved oxygen (80–100% saturation). For biosecurity, the AGD-affected fish (*n* = 20) were held in a 1000 L seawater tank with a flow rate of ~10 L/min linked to a fully integrated recirculation system complete with particulate filtration (~100 µm), 2 × 0.75 kW drive pumps, a 20 L protein skimmer and a trickle biofilter (100 L). Critical water quality parameters were maintained including salinity (35 ppt), pH (7.8), dissolved oxygen (80–100%) and temperature (16 ± 1 °C). Both systems were equipped with inbuilt monitoring alarms for water temperature and dissolved oxygen, as well as automated release of oxygen for emergency situations. All fish were fed a commercial dry pellet (Spectra SS 4 mm, Skretting Ltd., Australia) once daily to satiety.

Fish were inoculated with 500 cells/L of *N. perurans* the causative agent for AGD, as previously described by Taylor et al. [16]. Visual observation at 21 days confirmed normal AGD progression. AGD-affected fish were assessed according to a six-point standard gill scoring index [17] to be either a gill score of 3 (*n* = 8) or 4 (*n* = 2). Control and AGD-affected fish were humanely killed in 100 mg/L AQUI-S (AQUI-S New Zealand Ltd., Lower Hutt, New Zealand). Swabs were collected from three of the remaining AGD-affected fish and microscopy was undertaken to confirm the morphology was consistent with *N. perurans*. Gill baskets were dissected and split across multiple fixatives including seawater Davidson’s (2:3:1:3 parts of 37% formaldehyde (Sigma-Aldrich 252549), ethanol, glacial acetic acid, and seawater), Carnoy’s (6:3:1 parts of ethanol, chloroform and glacial acetic acid) and Neutral Buffered Formalin (Sigma-Aldrich HT501128). Gills for mucin extraction were immediately frozen in extraction buffer (6 M guanidine hydrochloride (GuHCl), 5 mM EDTA, 10 mM sodium phosphate buffer (pH 6.5), 0.1 mM phenylmethylsulfonyl fluoride (PMSF)).

### 2.3. Mucin Extraction

For the ATS-SE, mucins were extracted and isolated using isopycnic density gradient centrifugation as previously described [4]. Mucins were extracted from ATS-AU by mixing tissue with two times the sample volume of extraction buffer and subjecting samples to four strokes with a loose pestle using a Dounce homogenizer, followed by incubating the tissue slurries on a rocking board for 20 h at 4 °C, followed by centrifugation at 3900× g for 80 min. The supernatants containing the mucins were kept at 4 °C. We have previously demonstrated that these two methods of mucin isolation yield similar *O*-glycan profiles using liquid-chromatography–mass spectrometry (LC-MS), with differences between *O*-glycan profiles being of a similar magnitude to those observed among technical replicates within each extraction method [3].

### 2.4. O-glycan Analysis by Liquid-Chromatography–Mass Spectrometry (LC-MS)

Mucins (approximately 100 μg) were dot-blotted on PVDF membrane (Immobilon P membranes, Millipore, Billerica, MA, USA). The dots were visualized by Alcian blue (AB, Sigma-Aldrich). Blue dots were excised and subjected to reductive β-elimination. The dots were treated with 50 mM NaOH and 1 M NaBH_4_ for 16 h at 50 °C and the reaction was then quenched by addition of 2 µL glacial acetic acid. An AG50WX8 cation-exchange resin (Bio-Rad) was used to desalt the samples in a reverse-phase μ-C18 ZipTip (Millipore). Eluates were dried in a SpeedVac concentrator and borate was removed by evaporating added methanol five times. Released *O*-glycans were analyzed by LC-MS using a 10 cm long, 250 μm inner diameter column prepared in-house, containing 5 μm porous graphitized carbon (PGC) particles (Thermo Scientific, Waltham, MA, USA), as previously described [3]. A linear LC gradient of 0–40% acetonitrile in 10 mM ammonium bicarbonate, at a flow rate of ~10 μL/min, was used to elute the *O*-glycans. An LTQ mass spectrometer (Thermo Scientific) in negative-ion mode with an electrospray voltage of 3.5 kV, capillary voltage of −33.0 V, a capillary temperature of 300 °C with compressed air as a sheath gas was used to detect the *O*-glycans. Full scans were performed in the mass range of *m*/*z* 380–2000 in LC-MS. MS/MS was performed at a normalized collisional energy of 35% with a minimal signal of 300 counts, isolation width of 2.0 *m*/*z*, and an activation time of 30 ms. The data were processed using Xcalibur software (version 2.0.7, Thermo Scientific). Glycans were manually annotated from their MS/MS spectra with the help of Glycoworkbench 2 software [18] and validated using the UniCarb-DB database [19] or previous studies if the structure had been previously identified [3,4]. The following assumptions were made during structural annotation: the monosaccharide at the reducing end was assumed to be GalNAcol, hexoses were assumed to be Gal residues and *O*-glycans with linear cores (core 1, 3 and 5) were differentiated from branched cores (core 2) based on the presence of [M-H]^−^-223 and [M-H]^−^-C_3_H_8_O_4_ (-108) in the MS/MS of structures with linear cores [4,20]. Epitope-specific fragmentation and biosynthetic pathways were also used to identify structures. The relative amount of glycans were calculated based on the precursor mass specific peak areas in the LC-MS chromatograms. The relative amount of each glycan was calculated as a percentage of the total area of all selected peaks. The Symbol Nomenclature for Glycomics (SNFG) were used to represent proposed structures [21].

### 2.5. Fucosidase Treatment

Two samples of purified *O*-glycans from ATS gills were tested with fucosidase to potentially see differences in structures after running LC-MS. By digesting the fucosylated structures by the fucosidase, fucose linked to GlcNAc should be decreased in greater amounts than fucose linked to GalNAc, since the enzyme digest Lewis structures, which have fucosylated GlcNAc [22]. The samples were added to 19 µL of the supplied buffer and 1 µL (1 mU) Fucosidase α1-2, 3, 4, 6 (New England Biolabs Inc, Ipswich, UK) and put in 37 °C for 24 h. Then, the samples were purified with porous graphitized carbon in a μ-C18 ZipTip (Millipore), before running LC-MS.

### 2.6. Ex Vivo Imaging

Prior to imaging, individual gills from ATS-SE were immobilized by pinning them to silicone gel dissection plates using needles. For low-magnification imaging of mucus coverage, gills were immersed in Ringer’s solution for fresh water-acclimated salmonids (Ringers-FW: 140 mM NaCl, 2.5 mM KCl, 1.5 mM CaCl_2_, 0.8 MgSO_4_ and 5 mM HEPES and pH set to 7.8 with 1.5 M TRIS-base) solution containing yellow-green fluorescent 0.2 µm diameter microspheres (ThermoFisher, F8811) at a 1:20 dilution. Gills were incubated for 10 min in the dark at room temperature to allow microspheres to become trapped in mucus, and then gently washed with Ringers-FW solution. Brightfield and fluorescence photomicrographs were acquired using an SMZ18 stereomicroscope. For high-magnification imaging of gill mucus and assessment of mucus barrier properties, gills were first immersed in a Ringers-FW staining solution containing 20 µg/mL each of the fluorophore-conjugated lectin *Sambucus nigra* (SNA)-Cy3 (CL-1303, Vector Laboratories, Burlingame, CA, United States) and *Aleuria Aurantia* Lectin (Vector Laboratories, L-1390), conjugated to the fluorophore Alexa Fluor 647 using the Alexa Fluor™ 647 Protein Labeling Kit (ThermoFisher, A20173). Bacteria-sized (1 µm) blue fluorescent microspheres (ThermoFisher, F8814) were added to the staining solution at a 1:20 dilution. Gills were incubated with the staining/bead solution for 10 min in the dark at room temperature, then gently washed with Ringers-FW. Stained gills, mucus and beads were visualized by generating confocal z-stacks using a LSM 700 microscope equipped with a 20× water immersion objective (Zeiss, Baden-Württemberg, Germany).

### 2.7. Morphology and Histology

Gross gill pathology was documented by photography of fixed tissue. The gill arches were transferred to 70% ethanol within 24 to 48 h and then dehydrated and set in paraffin wax within a week after fixation. The gill arches from each fish were sectioned at 5 µm and stained with hematoxylin and eosin. All sections were viewed under an Axio Imager M2 microscope and imaged with an Axiocam 506 Color.

Sections (7 μm) from paraffin blocks containing Carnoy fixed tissue were stained with PAS/Alcian blue (PAS/AB, pH 2.5 [4]) to quantify the ratios of neutral (pink) vs. acidic (blue) mucins. Areas of the gill with healthy morphology as well as regions showing lamellar epithelial hypertrophy were selected from fish with AGD and compared with the gills from the non-infected fish. From each individual (*n* = 20), a minimum of 30 goblet cells (or sites in secreted mucus) were sampled for their RGB color codes in ImageJ software. The red/blue ratio was calculated for each of these data points, the means of which were used as biological replicates yielding *n* = 10 in each group.

### 2.8. Statistics

Cluster 3.0 with spearman rank correlation and centroid linkage was used to perform hierarchical clustering and Java Treeview (version 1.1.6r4, Santa Clara, CA, USA) was used to visualize the cluster [23]. GraphPad Prism 7.0 (GraphPad Software Inc., San Diego, CA, USA) was used to compare groups with the Mann–Whitney U-test and Kruskal–Wallis test with Dunn’s corrections for multiple comparisons. Red/blue ratio of PAS/AB staining were analyzed with one-way analysis of variance (ANOVA) with Dunnett´s post hoc test. Data analyzed with one-way ANOVA or presented as means + standard error of the mean (SEM) passed the Shapiro–Wilk test of normality. *p*-values < 0.05 were considered significant.

## 3. Results

### 3.1. ATS-SE Gill Mucin Glycans Are More Fucosylated and Less Sialylated Compared to Skin and Intestinal Mucin Glycans

We compared gill mucin O-glycosylation from ATS-SE with that of skin and gastrointestinal tract (GI: pyloric caeca and distal intestine) using LC-MS. We found 47 gill mucin *O*-glycans (Appendix A), whereof eight structures were common to both skin and GI (Figure 1A). The gill glycans were larger, more complex and exhibited a wider range of structures than the previously published skin mucins from the same fish [3]. The gill *O*-glycans were distinctly different from that of the skin glycans, but more similar to skin glycans than to GI glycans (Figure 1A,B). A major distinguishing feature of gill glycans compared to glycans from the skin and GI was the high abundance of terminal *N*-acetylhexoseamines and fucose, at the expense of a lower proportion of sialylated structures (Figure 1C).

### 3.2. Gill Mucin O-glycans Carry Fucosylated Structures Similar to the Mammalian Histo-Blood-Group Antigens

Fucosylated structures often have important immunological significance, such as the ABO blood group system in humans, however this has not been explored in fish. By examining the identified fucosylated structures in the gill mucin *O*-glycans, three types of fucosylated epitopes were elucidated: Fuc-HexNAc-R, Gal-[Fuc-]HexNAc-R and HexNAc-[Fuc-]HexNAc-R. The gill mucins contain more of these fucosylated structures than skin mucins (Figure 2A). The fucosylated structures were identified from MS/MS distinctive losses of 214 Da, indicating a –Fuc-H_2_O-CH_2_O ring-cleavage fragment, seen for HexNAc-linked fucose [4] (Figure 2B–D). Fucα1-3GalNAc has been found in zebrafish [24], which together with testing two of the samples by digestion of the *O*-glycans with α1-2, 3, 4, 6 fucosidase and then running LC-MS, suggests that the Fuc-HexNAc is mainly Fucα1-3GalNAc with a small portion being Fucα1-3GlcNAc. The percentage of digestion of Fuc-HexNAc terminals were 27% and 43% in the tested samples, while HexNAc-(Fuc-)HexNAc terminals were not digested (Appendix A). Fucosidase has been shown to digest Lewis-type structures (Fuc linked to GlcNAc) [22]. However, since most structures show resistance towards digestion, this suggests that the majority of the structures are Fuc-GalNAc. To our knowledge, this is the first report of terminal fucose-linked GlcNAcs described in the literature.

### 3.3. Fucosylated Mucins Appear to Coat Sialylated Mucin Strands in Gill Mucus

Fluorescent beads (0.2 μm) which penetrate and become trapped in the mucus were first used to visualize the overall structure and location of gill mucus. Mucus was mainly present at the filament tips, but patchy mucus could also be seen overlaying the gills and between gill filaments from ATS-SE (Figure 3). Mucus with a strand- or bundle-like structure was especially evident at the filament tips (Figure 3B,E). To investigate if mucin glycosylation was differentially located among the mucus, *Sambucus nigra* lectin (SNA) was used to visualize the dominant sialic acid among gill mucins (α-2,6 linked sialic acid) and *Aleuria aurantia* lectin (AAL) was used to visualize fucosylated glycans. Sialylated mucins were present throughout the mucus bundles/strands, whereas fucosylated mucins appeared to cover the outside of the sialylated mucus bundles (Figure 4). Bacteria-sized beads (1 μm) did not penetrate the mucus (Figure 4), however, since the gill mucus was discontinuous, it does not appear to have the thick barrier function against pathogens that is present, for example, in the mouse colon [25,26].

### 3.4. Gill O-Glycans were Larger, More Fucosylated and Less Sialylated Compared to Skin O-glycans both among ATS-SE and ATS-AU

Skin and gastrointestinal ATS *O*-glycosylation has previously been demonstrated to differ between geographical regions [3]. From the mass spectrometry data, the structure at *m/z* 716 (GalNAcα1-3[NeuAcα2-6]GalNAcol) was the most dominating structure among both the Swedish and Australian salmon gills (Appendix A). However, sialylated structures such as *m/z* 513, 675a, 675b and 966 had a higher relative abundance in the ATS-AU, and fucosylated structures such as *m/z* 571b, 733a, 936a and 1854a had a higher relative abundance in the ATS-SE (see Appendix A). However, in spite of differences in strain, salinity, husbandry and harvesting, the gill glycans were larger, more fucosylated and less sialylated compared to skin glycans in both ATS-SE and ATS-AU (Figure 5).

### 3.5. AGD Alters Gill Glycosylation in ATS-AU

To elucidate the effect of AGD on gill glycosylation, we then studied the mucin *O*-glycans from gill swabs from 10 control and 10 ATS-AU with AGD. Seawater-adapted ATS-AU were challenged with *N. perurans* to induce gross lesions in all infected fish, with gill scores at sampling observed to range from 3 to 4 (average 3.2). Corresponding histological lesions were observed in the AGD-affected fish and were characterized by extensive lamellar fusion and epithelial hyperplasia typical of AGD (Figure 6B). Trophozoites were observed associated with the margins of hyperplastic lesions (Figure 6B). The histology of the control fish was normal (Figure 6A) and the weight range was similar between infected and uninfected fish (Figure 6C).

PAS/AB staining of *N. perurans*-infected ATS gills resulted in a higher proportion of red- vs. blue-stained secreted mucus and goblet cells in areas with lamellar epithelial hypertrophy compared to gill areas with healthy lamellar and filament morphology in infected fish (*p* ≤ 0.05; *n* = 20) as well as with gills from the uninfected group (*p* ≤ 0.001; *n* = 20, Figure 6D). There was also a tendency for higher red/blue ratio in morphologically healthy-looking gill areas of infected fish compared to uninfected fish (Figure 6D). Since PAS/AB stains acidic mucins blue and neutral mucins pink/red, these results suggest that glycosylation changes occur in response to the infection.

A total of 88 *O*-glycans were identified using LC-MS, whereof 82 were present in both groups (Figure 7A, Appendix A). The majority of the glycans were composed of two monosaccharides but structures comprised by up to 12 monosaccharides were detected (Figure 7B). Structures composed of core 1, 2, 3 and 5 were found, with core 5 being the most common with an average of ~40% relative abundance in both infected and uninfected groups. Sialyl-Tn were included in the calculation and had a similar abundance as core 5. Core 1 structures were less abundant among infected fish compared to uninfected fish (Figure 7C).

HexNAcs and NeuAc were the most abundant terminal residues with a relative abundance of ~60% each, followed by fucose at ~30%. Structures with acidic moieties such as NeuAc and sulfate groups were significantly less abundant, whereas the neutral terminal moieties HexNAcs and fucose tended (*p* = 0.166 and *p* = 0.089) to be more abundant among AGD-affected fish (Figure 7D). The 30 most abundant structures are shown in Figure 8. The structures identified were generally short sialylated structures such as *m/z* 513, 675a, 675b, 716 and 966 in both infected and uninfected fish. On average, the uninfected group had more of these sialylated structures. The core 5 structure at *m/z* 425 was also prominent in both cohorts. No major differences were found among the cohorts, but the sulfated *m/z* 755 structure was more abundant in uninfected fish while long fucosylated structures such as *m/z* 1139b, 1139cc and 1447 were expressed more in infected fish. The structures at m/z 587 and 790 also had a higher relative abundance in infected fish (Figure 8).

The fucosylated and NeuAc-containing *O*-glycans were inversely proportional, with infected fish on the lower scale of NeuAc abundance and high on fucosylated structures (Figure 9A). The Lewis Gal-[Fuc-]HexNAc-R epitope constituted a very small part of the fucosylated structures, while Fuc-HexNAc-R were expressed in all individuals to varying degrees. However, the HexNAc-[Fuc-]HexNAc-R epitope was abundant in only four of the uninfected and five of the infected samples, with small traces observed in the other individuals (Figure 9B).

## 4. Discussion

We have identified that gill mucin glycans are larger, more complex and carry a wider range of structures than skin mucin glycans. The gill mucin glycans are more fucosylated and less sialylated compared to skin and intestinal mucin glycans. The fucosylated structures on gill mucins are similar to the mammalian histo-blood-group antigens, leading to greater structural diversity between individuals compared to mucin glycans from other organs. In ex vivo gill mucus, fucosylated mucins appear to coat sialylated mucin strands. Finally, we demonstrated that mucin *O*-glycans from AGD-affected ATS-AU differ from un-affected ATS-AU, especially with regards to a decrease in sialylated and sulfated structures and an increase in structures carrying HexNAc among AGD-affected fish.

It is tempting to use skin mucus as a proxy for gill mucus due to non-invasive tissue sampling. However, gill mucin glycans were distinctly different from skin mucins from the same fish, with regards to size, complexity, charge, terminal moieties and structural diversity. Skin mucins can thus not be used as a proxy for gill mucins in general. With that said, it is still possible that specific potential biomarkers, either among the mucin glycans or among other mucus components, may be present in both skin and gill mucus.

Regardless of geographical origin, strain and environmental salinity, the gill mucins carried a larger proportion of fucosylated *O*-glycans than skin and intestinal mucins. The fucosylated epitopes were of three distinguishable types: Fuc-HexNAc-R, Gal-[Fuc-]HexNAc-R and HexNAc-[Fuc-]HexNAc-R. The Lewis Gal-[Fuc-]HexNAc-R epitope constituted a very small part of these, while Fuc-HexNAc-R were expressed in all individuals to varying degrees. However, the HexNAc-[Fuc-]HexNAc-R epitope was abundant in only four of the uninfected and five of the infected samples, with small traces observed in the other individuals. This could potentially be a blood group epitope, which can selectively be expressed as *O*-glycans on certain epithelia as histo-blood-group antigens, however further studies are needed to verify this. Little is known about carbohydrate species and blood groups in fish. In the 1960s, blood group systems with 15 phenotypes in skipjack tuna (*Katsuwonus pelamis*) [27], four phenotypes in brown trout (*Salmo trutta*) and three phenotypes in rainbow trout (*Oncorhynchus mykiss*) [28] were described. However, although immunogenic reactions were identified, they did not conclude the cause of these or whether these blood groups are composed of carbohydrates or proteins. Fucα1-3GalNAc has been found in zebrafish (*Danio rerio*) [24], which together with our results from digestion of *O*-glycans with α1-2, 3, 4, 6 fucosidase, suggests that the Fuc-HexNAc is mainly Fucα1-3GalNAc with a small portion being Fucα1-3GlcNAc, and the HexNAc-[Fuc-]HexNAc is hard to identify, but is speculated to be based on Fuc-GalNAc since it shows resistance against the fucosidase and shows higher relative abundance in samples with lower relative abundance of Fuc-GalNAc, suggesting it is the precursor. The relevance of these structures is still unexplored and requires further investigation. Considering that the inter-individual diversity of mucin glycans is extremely low compared to that of mammals and glycan diversity in mammals is considered a way to limit the risk of wiping out an entire population by a single pathogen [3,4,5,6,7,29], these systems could be an avenue to diversify the glycan repertoire in the vital and exposed gill. However, they could also have other functions, especially since they are similar to Lewis epitopes that have been demonstrated to have important roles in immunology and infection in mammalian epithelia [30,31].

The mucosal barriers of the gill, skin and gut are the first line of defense for teleosts. Similar to other vertebrates, the mucus of teleost fish contains a number immunologically important factors, including immunoglobins, cytokines, proteases, lysozyme, antimicrobial peptides and complement factors. Previous studies have shown that a number of these proteins are differentially regulated in response to AGD [32]. However, the mucins form a gel which keeps these other molecules in place, and changes to the mucus gel may affect its ability to keep these molecules in a strategic position. The mucins are mainly comprised of carbohydrates, which themselves can affect pathogen growth, virulence and adhesion to epithelia [5,33,34,35]. Furthermore, in mammals, it has been demonstrated that gut glycans govern the microbiota and can have roles in both gut-centered and systemic infection and inflammation [36].

In line with the results of increased PAS to AB stain ratio in areas with lamellar epithelial hypertrophy in the AGD-affected fish in our study, others have described tendencies for increased levels of PAS-positive mucins in regions affected by lesions [14]. Such changes suggest that glycosylation changes occur with infection, and indeed we detected these using LC-MS. The abundance of 6 of the 30 most abundant structures differed significantly between infected and uninfected fish, and neutral glycans increased whereas charged glycans decreased, reflecting the results from the PAS/AB-stained tissue. We cannot exclude that unknown factors that have potential to affect glycosylation (i.e., microbiota) might differ between the control and AGD tanks. However, the shift from acidic to neutral glycans has been corroborated in other studies [14], suggesting that the glycans identified to differ between AGD-affected and control ATS-AU are most likely caused by AGD.

In the intact gill, mucus was mainly present at the filament tips in structures resembling bundles or strands, but patchy mucus could also be seen overlaying the gills and between gill filaments. Bacteria-sized beads did not penetrate the mucus, however, since the gill mucus was discontinuous, it does not appear to have the thick barrier function against pathogens that is present, for example, in the mouse colon [25,26]. Sialylated mucins were present throughout the mucus bundles/strands, whereas fucosylated mucins appeared to cover the outside of the sialylated mucus bundles. Some mucins carry both sialylated and fucosylated structures, so the difference between sialylated and fucosylated mucins is not entirely distinct, but more likely a density/concentration difference where the sensitivity of the lectin stains used for visualization lead to a low signal in places where the epitopes are present at low concentrations. Gills may have a clearance system similar to the pig airways in which MUC5B mucin bundles coated with MUC5AC trap and remove particles [37]. It is possible that the change in the relationship between fucosylated and sialylated mucins may affect such clearance systems. We also identified that the proportion of large glycans was higher among mucin glycans from AGD-affected gills than unaffected gills. Extended glycans also allows the terminal structures to be more accessible to microorganisms, which may also affect the clearance ability of the mucus. However, both the clearance system per se and effects of glycosylation changes on its efficiency requires further investigation before any firm conclusions can be drawn.

Increases in mucus cell numbers at the distal tips of non-hyperplastic secondary lamellae have been described in naturally AGD-affected ATS and increases in skin mucus cell numbers and decreases in mucus viscosity have also been found in salmonids affected by AGD compared to uninfected fish [14,15]. These observations have been interpreted as an increase in mucus production, although mucus production (as opposed to hydration and composition) during AGD was not explicitly measured. Another study found increased expression of *Muc5* and genes previously shown in to be involved in mucus secretion in mammals in AGD-affected gills [38]. In the present study, we identified that the proportion of large glycans was higher among mucin glycans from infected gills than from uninfected gills. In the murine intestine, increased glycan size has been shown to be associated with decreased mucin biosynthesis rate due to longer time spent in the intracellular biosynthesis machinery [39]. Considering that mucus formation depends on multiple aspects, including mucin apoprotein production, glycosylation, secretion and ion composition [25,40,41], an alternative interpretation is therefore that these changes are caused by physiological changes and a decreased mucus turnover. Due to the assumption that AGD increases mucus production, a number of studies have examined the effects of mucolytic agents on the onset and progression of AGD. L-cysteine ethyl ester and *N*-acetyl cysteine reduced salmon mucus viscosity and dietary L-cysteine ethyl ester delayed AGD onset, whereas *N*-acetyl cysteine was not a successful AGD treatment [15,42,43]. Considering that these compounds have effects additional to mucolytic effects, including antioxidant and antimicrobial effects [44], these contradicting results might be unrelated to their shared mucolytic effects. Regardless, considering the large size of this pathogen and the discontinuous gill mucus suggested by the current study, the parasite may invade through this defense mechanism and cause extensive tissue damage.

In conclusion, care must be taken when using skin mucus as a proxy for gill mucus, as gill mucins are distinctly different from skin mucins with regards to size, complexity, charge, terminal moieties and structural diversity. Fucosylated epitopes of three distinguishable types with similarities to histo-blood group antigens induce increased inter-individual diversity in gill mucins compared to mucins from other epithelia. This may limit the risk of infection in the vital and exposed gill. However, since they are similar to Lewis epitopes, which have important roles in immunology and infection in mammalian epithelia, they could also have other functions. Furthermore, fucosylated mucins appeared to cover the outside of the sialylated mucus bundles, and infection-induced changes in the ratio between fucosylated and sialylated mucin glycans may therefore affect the efficiency of the clearance system.

## Figures and Tables

**Figure 1 microorganisms-08-01871-f001:**
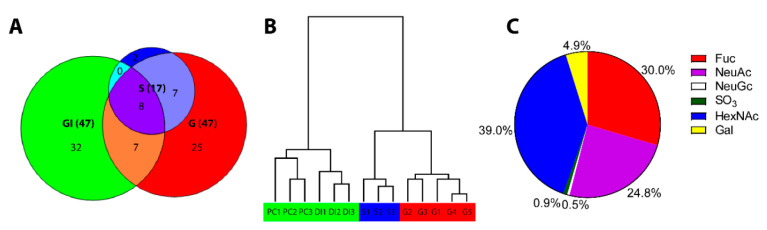
Characterization of Swedish gill mucin *O*-glycans and comparison with *O*-glycans from other epithelia in the same fish. (**A**) Venn diagram of detected *O*-glycan structures in skin (S), gill (G) and gastrointestinal (GI) mucins. (**B**) Tree view of hierarchical clustering of *O*-glycan profiles from pyloric caeca (PC), distal intestine (DI) and skin (S). The detailed characterization of the glycans from non-gill epithelia has been published previously [3]. (**C**) Pie chart of the percentage of terminal ends occupied by respective moiety among gill mucins.

**Figure 2 microorganisms-08-01871-f002:**
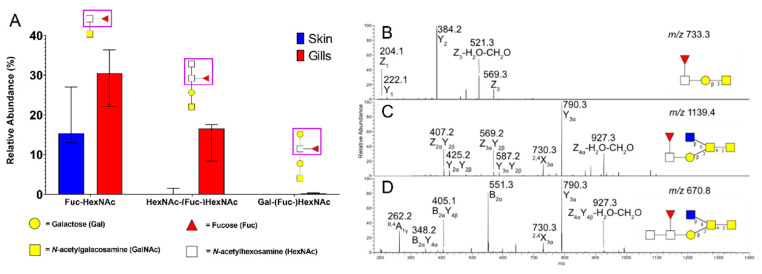
Description, identification and characterization of fucosylated structures. (**A**) Comparison of fucosylated structures in ATS-SE (Swedish ATS) skin and gills. Bars display median with interquartile range. (**B**) *m/z* 733.3, retention time (rt): 13.1 min, (**C**) *m/z* 1139.4, rt: 15.0 min, (**D**) *m/z* 970.8 ([M-2H^+^]^2−^), rt: 16.0 min.

**Figure 3 microorganisms-08-01871-f003:**
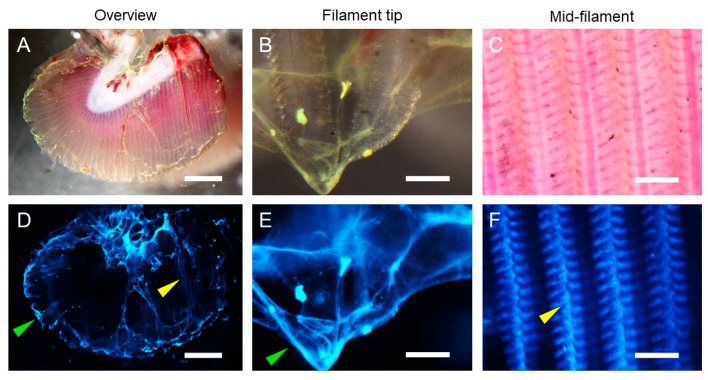
Ex vivo imaging of 0.2 µm fluorescent beads trapped in gill mucus. Representative micrographs showing gill overview (**A**,**D**) and higher magnification images from the gill filament tip (**B**,**E**) and mid-filament regions (**C**,**F**) from ATS-SE. Beads trapped in mucus bundles at the gill filament tip (green arrows) and between the gill filaments (yellow arrows) are indicated. Upper panels (**A**–**C**) show brightfield images and equivalent fluorescence images (**D**–**F**) are shown in lower panels. Images are representative of *n* = 5 animals; scale bars are 1 mm (**A**,**D**) and 100 µm (**B**,**C**,**E**,**F**).

**Figure 4 microorganisms-08-01871-f004:**
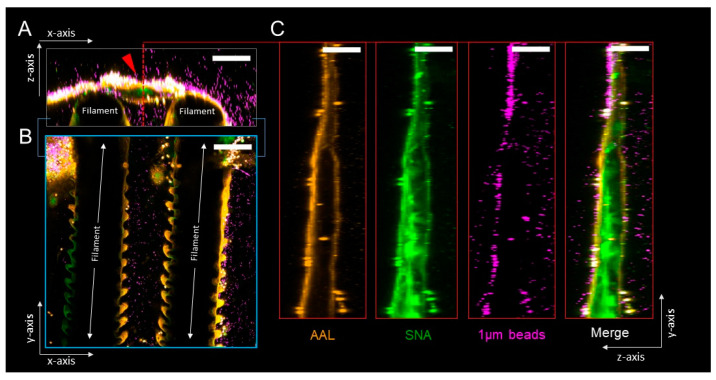
Ex vivo imaging of gill mucus glycosylation and barrier properties. Representative micrographs showing confocal z-stack cross sections of ATS-SE gills stained using fluorophore-conjugated AAL (*Aleuria aurantia* lectin, detects fucose, orange) and SNA (*Sambucus nigra*, detects sialic acids, green) lectins, and overlaid with bacteria-sized 1 µm fluorescent beads (purple). (**A**) x/*z*-axis cross-section through mucus bundle (red arrow) positioned between two gill filaments. (**B**) x/*y*-axis cross-section through gill filaments shown in (**A**). (**C**) y/*z*-axis cross-sections through mucus bundle indicated by red dashed line in (**A**); panels show individual and merged fluorescent signals. Images are representative of *n* = 5 animals; scale bars are 100 µm.

**Figure 5 microorganisms-08-01871-f005:**
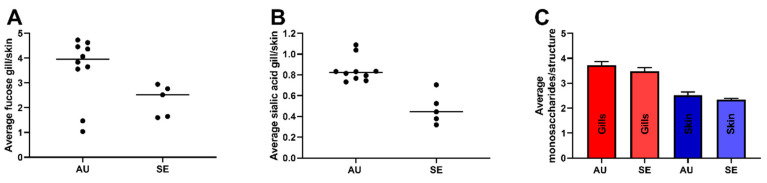
Gill *O*-glycans were more fucosylated, less sialylated and larger compared to skin *O*-glycans, regardless of origin. (**A**) Fold difference in average number of fucose residues per glycan in gill compared to skin mucin *O*-glycans in Australian (AU) and Swedish (SE) ATS. ATS-AU skin mucin *O*-glycans have been described in detail previously [3]. Bars show the median. (**B**) Fold difference in average number of NeuAc (*N*-acetylneuraminic acid) residues per glycan in gill compared to skin mucin *O*-glycans. Bars show the median. (**C**) The average number of monosaccharide residues per glycan, Mean + standard error of the mean (SEM).

**Figure 6 microorganisms-08-01871-f006:**
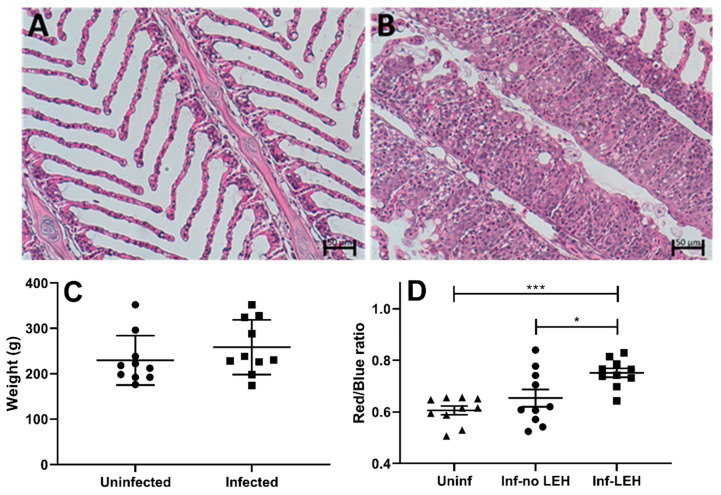
Disease parameters and histological characteristics in uninfected and *N. perurans*-infected gills. (**A**) Gill histology of uninfected ATS-AU showing normal filament and lamellae structure. (**B**) Gill histology of AGD-affected ATS demonstrates extensive hyperplasia of epithelium. *N. perurans* trophozoites are clearly visible on the margins between hyperplastic lesions. (**C**) Weight of uninfected and *N.perurans*-infected fish. (**D**) Red/blue ratio of PAS/AB secreted mucus and goblet cells of ATS. *N. perurans*-infected ATS gills had a tendency for more pink staining in healthy-looking areas (no LEH) as compared to that of uninfected fish (*p* = ns). The pink staining was significantly stronger in LEH regions of infected fish compared to both uninfected fish (*p* ≤ 0.001; *n* = 20) and healthy-looking regions of the same infected fish (“Infected, no LEH”; *p* ≤ 0.05; *n* = 20). Statistics: one-way analysis of variance (ANOVA) with Dunnett´s post hoc test against the uninfected group displayed by mean + SEM; * *p* ≤ 0.05; *** *p* ≤ 0.001. Abbreviations: LEH = lamellar epithelial hypertrophy, uninf = uninfected, inf = infected.

**Figure 7 microorganisms-08-01871-f007:**
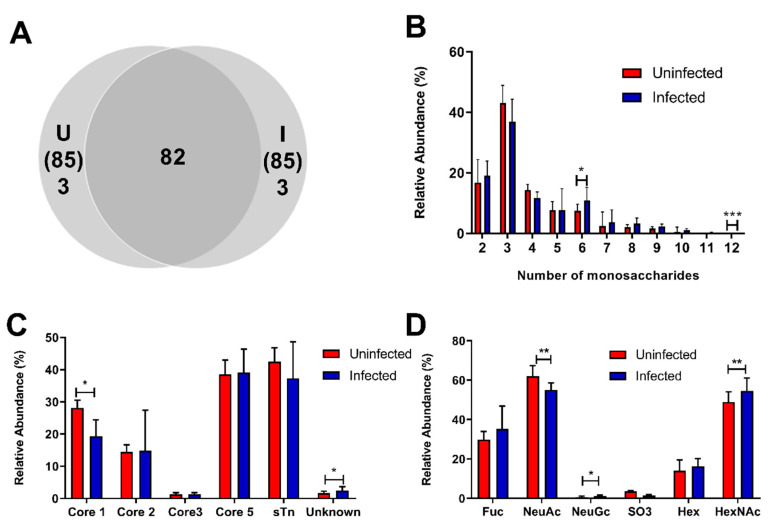
Overview of *O*-glycan characteristics in uninfected and *N. perurans*-infected gills. (**A**) Venn diagram of number of identified structures in the uninfected (U) and AGD-affected (I) groups. (**B**) Size distribution of the glycans, i.e., the number of monosaccharides in the structures. (**C**) Distribution of core structures. (**D**) Relative abundance of structures with corresponding terminal structures. Bars show the median with interquartile range and the significance was calculated using the Mann–Whitney U-test. **p* < 0.05, ***p* < 0.01, ****p* < 0.001.

**Figure 8 microorganisms-08-01871-f008:**
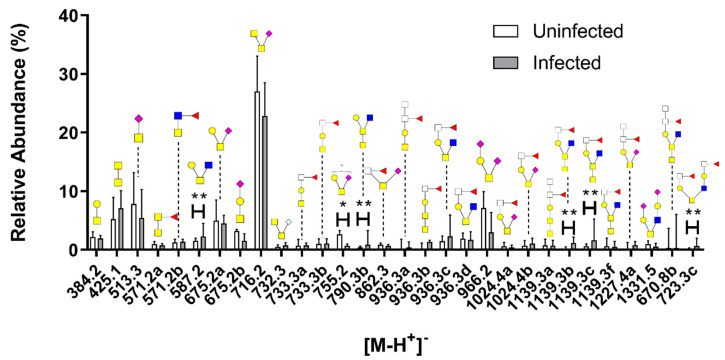
Comparison of the 30 most abundant *O*-glycan structures from uninfected and *N. perurans*-infected gills. The figure displays the median with interquartile range and significances were calculated with the Mann–Whitney U test. * *p* < 0.05, ***p* < 0.01. Symbols: red triangle = Fuc, yellow circle = Gal, blue square = GlcNAc, yellow square = GalNAc, white square = HexNAc, purple diamond shape = NeuAc and S = sulfate group.

**Figure 9 microorganisms-08-01871-f009:**
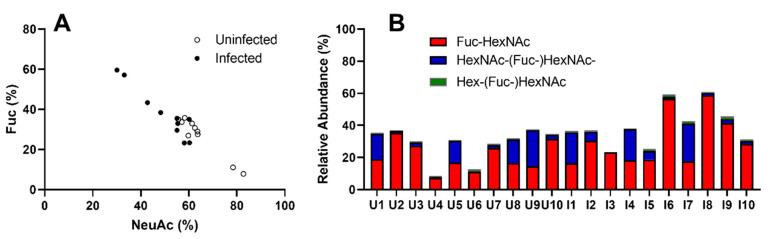
Fucosylated structures and their expression in *N. perurans*-infected and uninfected fish. (**A**) The relative abundance of fucosylated and sialylated structures in individual fish were plotted for comparison. The levels of fucosylated and sialylated structures are inversely proportional. (**B**) Graph displays the fucosylated structures separated into three groups based on terminal epitope. Seven individuals express substantial levels of HexNAc-(Fuc-)HexNAc epitopes (13–24%), while another two a moderate amount (~5%). U = uninfected, I = infected.

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
