# Peer review of "Gill Mucus and Gill Mucin O-glycosylation in Healthy and Amebic Gill Disease-Affected Atlantic Salmon"

_microorganisms, 2020, doi:10.3390/microorganisms8121871_

Round 1
Reviewer 1 Report
This manuscript is intersting and offers some insight into the differences between infected and non-infected gills. It is also intersting that differences are apparent between different strains of salmon although these are perhaps not well reflected in the manuscript. This perhaps may reflect the origins of the salmon (Swedish being Baltic and Australian being North American). I am concerned that the comparisons made are not good ones - rather like trying to fit two different jigsaws together to make one picture. This would normally be a red flag to me and needs to be satisfactorily defended. Generally the manuscript is well written although structurally a bit awkward and needs to be clearer in the message that is being sent. For example starting the discussion with the conclusion (or a summary of the results) maybe helpful but if the results were more clearly written and explained, it is unnecessary. These sections, need to be integrated better into the body of the subsequent text. The following points need to be carefully considered and addressed.
Generic and species names need to italicised in the manuscript - they are often ignored and this is a basic scientific requirement.
L68 AGD infected - this is wrong it is a disease therefore not infected but AFFECTED
L69 Control fish scavenged? This wording does not seem appropriate. If these fish were from other experiments then this represents re-use of animals. In which case these should have had ethical approval.
L79 Why was metomidate used in Sweden but other anaesthetics in Australia - would/could this affected the results? Similarly the feed was different between fish groups!
L185-189 need to include better description of the statistics - were the data normal/homogeneity of variance? What was analysed with which tests?
L239-240 this is discussion not a result
L318 - no big differences - not appropriate scientific language
L328 and elsewhere - why the need to have 3 levels of significance - adds absolutely nothing and is meaningless with the data presented. it is significant or not at which ever level you choose - should have been defined at line 185-189.
Reviewer 2 Report
The present manuscript titled "Gill mucus and gill mucin O-glycosylation in healthy and amebic gill disease affected Atlantic salmon" presents nice work on the gill mucus during amebic gill disease (AGD). This work is from the nice collaborate work from experienced researchers. The manuscript has been well written with data analysis and presentation. The manuscript could be considered for publication.
Reviewer 3 Report
Experimental set-up and sampling of the AGD infection experiment:
-
The experimental procedures are not clear.
-
How many tanks were assigned for the AGD infection experiment?
- pH?
- How many fish were placed and sampled per tank?
- What Stocking density?
-
If the fish were fed ad libitum a recirculation system was necessary with biofilter
-
The techniques to isolate and to obtain amoebae for the experimental AGD infection are not described.
-
Has N. perurans been identified by molecular biology and morphological criteria?
- How many days were necessary after the inoculation period with amoebae to display the typical AGD-like lesions?
-
Are the fish re-exposed to the parasite at the same density?
- line 100: “assessed for AGD severity using a six point standard gill scoring method (16)”.Taylor et al. (2009 a, b) reported the gill score of AGD on a categorical scale of zero (no visible lesions) to five (advanced lesions covering the majority of the gill surface).
R.S. Taylor, W.J. Muller, M.T. Cook, P.D. Kube, N.G. Elliott (2009). Gill Observations in Atlantic salmon (Salmo salar L.) during repeated amoebic gill disease (AGD) field exposure and survival challenge. Aquaculture, 290, pp. 1-8
- The differences in farming salinity between ATS populations and the different anesthetic molecules used for euthanasia could have affected mucous quality and/or production?
- Write N. perurans in place of N. perurans
- Reduce the length of paragraph titles
- Discussion could be improved by citing other papers as Marcos-López, M., Calduch-Giner, J.A., Mirimin, L. et al. Gene expression analysis of Atlantic salmon gills reveals mucin 5 and interleukin 4/13 as key molecules during amoebic gill disease. Sci Rep 8, 13689 (2018). https://doi.org/10.1038/s41598-018-32019-8
Reviewer 4 Report
The paper of Benktander and colleagues describes an interesting aspect of mucosal physiological changes in AGD infected Atlantic salmon. They have studied the mucin O-glycosylation of healthy and AGD-infected salmon and found that gill mucin glycans were larger and far more complex and diverse than their counterparts in the skin. They further demonstrated that AGD infection could result in the O-glycans re-organisation and diversification, which were implicated to play a role in disease development.
The paper is well-written, the data are robust, and the implications drawn provide a fascinating avenue for future research on the gill mucosal physiology concerning AGD. The methods employed are innovative and top-notch. This paper offers a new perspective in AGD research, which is traditionally histology- and in recent years, transcriptomics-based.
It was not clear though how the gills scores of AGD infected fish were accounted for in the analysis. Can the authors comment on how the severity of infection (i.e. gill score) affect the gill O-glycomes? What was the parasitic load?
A congruence analysis would be beneficial to understand the relationship between gill score, glycomes, in both tissues.
The AGD infection trial needs additional information as in its present form, it is a bit unclear and confusing. The infection trial in 2.2. - this has been applied to the Australian trial, right? Were the ATS-SE also exposed to AGD? This should be clarified in the whole manuscript. I suggest that the authors separate the trials distinctively – The Swedish trial and The Australian trial – and clarify the analysis performed from the sample collected from each trial.
Though it is tempting to compare the SWE gills uninfected and AUS gills uninfected as shown in Figure 5, the authors need to be careful with their implications. The Swedish cohorts were reared at 2-3 ppt while the Australian group was at 33-35 ppt. I also assume the genetic background of the Swedish cohort was different from the Australian cohort. Moreover, the rearing temp for the Swedish experiment is missing. The authors should clarify and provide some information about these confounding factors that may influence the gill glycomes.
Minor:
- L90: smoltification is a more appropriate term for salmonid.
